# Perish and publish: Dynamics of biomedical publications by deceased authors

Chol-Hee Jung[1], Paul C. Boutros[2,3,4,5], Daniel J. Park[1,6], Niall M. Corcoran[7,8,9,10,11], Bernard J. Pope[1,7], Christopher M. Hovens[7,8,10]*

1 Melbourne Bioinformatics, The University of Melbourne, Melbourne, VIC, Australia, 2 Department of Human Genetics, University of California, Los Angeles, CA, United States of America, 3 Department of Urology, University of California, Los Angeles, CA, United States of America, 4 Jonsson Comprehensive Cancer Center, University of California, Los Angeles, CA, United States of America, 5 Institute for Precision Health, University of California, Los Angeles, CA, United States of America, 6 Department of Biochemistry and Pharmacology, University of Melbourne, Melbourne, VIC, Australia, 7 Department of Surgery, University of Melbourne, Royal Melbourne Hospital, Parkville, VIC, Australia, 8 Department of Urology, Royal Melbourne Hospital, Melbourne, VIC, Australia, 9 Department of Urology, Western Health, Footscray, VIC, Australia, 10 University of Melbourne Centre for Cancer Research, Victorian Comprehensive Cancer Centre, Parkville, VIC, Australia, 11 Victorian Comprehensive Cancer Centre, Parkville, VIC, Australia

* chovens@unimelb.edu.au

**Data Availability Statement:** All data will be available after acceptance of the manuscript. The data is now attached as S1 Table.

**Funding:** Support for our analyses was provided through a NHMRC project grant 1162514 to CMH

## Abstract

The question of whether it is appropriate to attribute authorship to deceased individuals of original studies in the biomedical literature is contentious. Authorship guidelines utilized by journals do not provide a clear consensus framework that is binding on those in the field. To guide and inform the implementation of authorship frameworks it would be useful to understand the extent of the practice in the scientific literature, but studies that have systematically quantified the prevalence of this phenomenon in the biomedical literature have not been performed to date. To address this issue, we quantified the prevalence of publications by deceased authors in the biomedical literature from the period 1990–2020. We screened 2,601,457 peer-reviewed papers from the full text Europe PubMed Central database. We applied natural language processing, stringent filtering and manual curation to identify a final set of 1,439 deceased authors. We then determined these authors published a total of 38,907 papers over their careers with 5,477 published after death. The number of deceased publications has been growing rapidly, a 146-fold increase since the year 2000. This rate of increase was still significant when accounting for the growing total number of publications and pool of authors. We found that more than 50% of deceased author papers were first submitted after the death of the author and that over 60% of these papers failed to acknowledge the deceased authors status. Most deceased authors published less than 10 papers after death but a small pool of 30 authors published significantly more. A pool of 266 authors published more than 90% of their total publications after death. Our analysis indicates that the attribution of deceased authorship in the literature is not an occasional occurrence but a burgeoning trend. A consensus framework to address authorship by deceased scientists is warranted.

and the PRECEPT program grant, co-funded by Movember and the Australian Federal Government to NMC. BP was supported by a Victorian Health and Medical Research Fellowship from the Department of Health and Human Services in the State of Victoria. NMC was supported by a David Bickart Clinician Researcher Fellowship from the Faculty of Medicine, Dentistry and Health Sciences, University of Melbourne, and more recently by a Movember – Distinguished Gentleman's Ride Clinician Scientist Award through the Prostate Cancer Foundation of Australia's Research Program. PCB is supported by NIH/NCI P30CA016042, 1U01CA214194-01 and 1U24CA248265-01.

**Competing interests:** The authors have declared that no competing interests exist.

## Introduction

The concept and definitions of what constitutes authorship in scientific publications has been much commented in the literature and numerous frameworks requiring self-checking by authors as to the fulfillment of these requirements have been adopted by journals and their editors over the last 40 years [1–6].

One of the most widely accepted reporting frameworks by journals in the medical literature field is that known as the ICMJE, the International Committee of Medical Journal Editors, which includes four criteria pertinent for authorship, all of which must be fulfilled in order to qualify for authorship. These clauses cover "1) Substantial contributions to the conception or design of the work; or the acquisition, analysis, or interpretation of data for the work; AND 2) Drafting the work or revising it critically for important intellectual content; AND 3) Final approval of the version to be published; AND 4) Agreement to be accountable for all aspects of the work in ensuring that questions related to the accuracy or integrity of any part of the work are appropriately investigated and resolved." [7].

Biomedical studies can take years to execute and publish. While substantial contributions can occur at any point during the conduct of a study, manuscript drafting tends to occur near its end and approval of the final manuscript and agreement to be accountable necessarily only occur near completion. There is thus an actuarial risk that some substantial contributors may pass away prior to final publication. While this risk is low for any given paper, millions of papers are published annually leading to a broad and recurrent issue: authorship for deceased individuals.

The ICMJE framework makes no specific reference to deceased authors and if these criteria were followed strictly, it would be impossible for any deceased author to fulfill all four criteria [8]. It may however be possible in some instances to fulfil most of these criteria. For example, an author who passes away between formal acceptance of a peer reviewed paper and its publication may meet the first three criteria. Nevertheless, such authors are manifestly unable to investigate or resolve any questions relating to the published work, and hence cannot meet criteria 4. In contrast, an individual who is already deceased prior to submission of a paper will be unlikely to be able to satisfy, in addition, criteria 2 and 3. Whether deceased individuals who have made substantial contributions to a study and therefore satisfy criteria 1, warrant inclusion as authors is a contentious issue that has elicited differing views in the literature [9–14]. Some biomedical journals, such as *BMJ Journal*s and *Pediatric Anesthesia* have now developed their own policies with respect to deceased authors. The BMJ Journals guide for authors requests that deceased authors deemed as appropriate as authors should be indicated as such with a death dagger and a footnote added indicating the deceased authors date of death [15]. The journal *Pediatric Anesthesia* has since 2016 instituted a new policy with regard to deceased authors. They have adopted the reporting requirements for deceased authors as listed in the BMJ Journals but in addition require an attestation from the corresponding author that all living authors agree that the deceased author has otherwise met the definition of authorship [12]. To date however there has not been a systematic analysis of the extent of deceased author contributions in the biomedical literature nor whether any trends in this practice are evident. To address these issues, we have stringently developed a database of deceased author publications in the biomedical literature since 1990. This has permitted a field wide analysis of the extent and changes over time of deceased author contributions in the biomedical literature and allowed exploration of the factors that might influence both the extent and trends in this practice.

## Material and methods

### Database search methodology

We utilized the programmatic interface of the Europe PMC full text database for automated retrieval of information on publications with deceased authors. The Europe PMC database (EPMC) was searched using the europepmcv0.4 R Client for the Europe PubMed Central RESTful Web Service [16] for all articles dating from 1990 to 2020 in the Acknowledgement section with the search terms 'deceased' OR 'perished' OR 'died' OR 'passed away' OR 'passed on' OR 'passed' OR 'RIP' OR 'passing' OR 'dedicate' OR 'dedicated' OR 'memory of'. The full text of 17,650 publications fulfilling these criteria were downloaded in XML format. Author status information can also be stored in the author information section of XML files. To ascertain how many records would contain this information, 356,812 full text XML files for all publications between 1995 to 2005 were extracted from EPMC, and PMC using a Python script. The author information section of the downloaded XML records were searched for the deceased status (deceased = 'yes'). This yielded only 16 publications by deceased authors, while our approach of analysing the Acknowledgement section found 83 publications for the same period. We therefore restricted our search for deceased authors to the Acknowledgement section of publications.

Sentences in the Acknowledgement section of the retrieved 17,650 publication were then analysed by spaCy, a Python package for natural language processing [17], to obtain linkages between the terms for death and an author name, filtered against the author field information from the same article. We limited the search results to contain a single author name to minimize the complexity in finding the linkages, hence any publications with potentially more than one deceased author were removed.

In total, 2002 publications were selected indicating the death of an author. Deceased authors with the same or similar name were disambiguated using other author information, such as affiliation, email, ORCID and co-author names. Complete manual inspection of the relevant sentence in the Acknowledgements removed false and incorrectly selected authors from this list of articles, leaving 1439 unique authors as the final set of deceased authors. The date of death of the deceased author was retrieved from the Acknowledgement or from obituaries. For those 1212 authors whose date of death was unavailable, the publication date of the earliest publication from which the deceased author was extracted was deemed as the latest possible date of death.

All publications by each of the unique set of 1439 deceased authors were searched for in the PubMed database using the person's name and affiliations, and the search results were downloaded in XML format. Each publication from the PubMed search was cross-checked with the information of the corresponding deceased author to exclude publications by authors with the same name. The publication date of each of the publications from the PubMed search and the date of death of the deceased author was compared at month level to decide whether a publication was posthumous. This analysis indicated that these 1439 deceased authors published a minimum total of 38,907 papers over their careers with 5,477 published after death.

To confirm that our automated retrieval script was not overestimating the number of publications for deceased authors, especially after death, we manually checked 599 authors, which included the top 125 ranked deceased authors, 399 bottom ranked authors and 75 randomly selected authors in the middle rank. For 546 out of these 599 authors (91.2%), the automatic retrieval script either underestimated or accurately assessed the true number of deceased publications compared to that confirmed by manual inspection. For 53 authors for whom the automatic retrieval script over-counted the posthumously published papers, the difference was 7 or less for 47 authors (86.8%). The difference for the remaining 6 authors was 11 or more with 66 being the biggest. These 6 authors had very common names, which only careful

manual inspection could confidently and conservatively disambiguate from other authors with the same name. We therefore manually curated all authors with common names and if they could not be readily disambiguated using affiliation linkages from other same name authors they were deleted from the list. Population estimates and crude death rates were extracted from 2019 Revision of World Population Prospects from the United Nations [18]. The estimated number of PubMed publications with deceased authors was calculated by factoring the proportion of deceased-author publications in EPMC for each year. The estimated number of deceased authors in PubMed was calculated similarly. We first calculated the percentage of deceased authors in EPMC, and then multiplied it to the pool of unique authors in PubMed.

## Odds ratios between the publications by the deceased authors after and before their death

For each of the 17 top journals, the odds of having publications after death were compared to that before death. For a journal, the odds-ratio was calculated from 2-by-2 contingency table using Fisher's exact test, which consists of the total number of publications by the deceased authors after death, the total number of publications in year 2019, the total number of publications by the deceased authors before death and, again, the total number of publications in year 2019. The p-value from the Fisher's exact tests were corrected for the multiple testing using the Benjamini & Hochberg method [19].

## Results

### The cumulative change in deceased author publications over time

We considered all publications dated 1990–2020 with open access full text available in Europe PubMed Central (EPMC, n = 2,601,457). We identified 17,650 articles that contained a term referencing a deceased individual (0.68% of 2,601,457 articles). After stringent filtering a set of 1,439 deceased authors contributing 1,756 articles was finalized (S1 Table). We found no publications from the period of 1990–1998 that mentioned a deceased author in the Acknowledgements section, with the first occurring in 1999 (Fig 1A).

To develop a career-wide view of these deceased authors, we then used the larger PubMed database. The number of publications including a deceased author has been growing rapidly. After the first appearance of a single paper in 1999, it rose to 6 in 2000, and increased to approximately 900 in 2019 –a 146 fold increase from the year 2000. This represents an annual 18.4% increase in the numbers of such publications since 2000. This significantly out-paces the growth rate in total PubMed publications of 2.5 fold for the same period, corresponding to an annual percentage increase of 5%. As a percentage of total publications, those including at least one deceased author grew from 0.001% in the year 2000 to 0.085% in the year 2018. The growth of total publications in the EPMC database is faster than that in PubMed, hence we adjusted the number of PubMed publications with deceased authors, by the proportional rate of such publications in EPMC. This derived an estimated rate of total publications by deceased authors in PubMed which is still significantly above the rate of total publications in PubMed (Fig 1A). Our use of EPMC to identify deceased authors and our strict filtering and conservative name-matching means that these values serve as a strict lower-bound for the numbers of publications by deceased authors.

### Factors associated with the change in deceased author publications over time

Many factors might underlie this apparent increase in the proportion of publications with deceased authors in the PubMed database from the years 2000 to 2019. One possibility is an

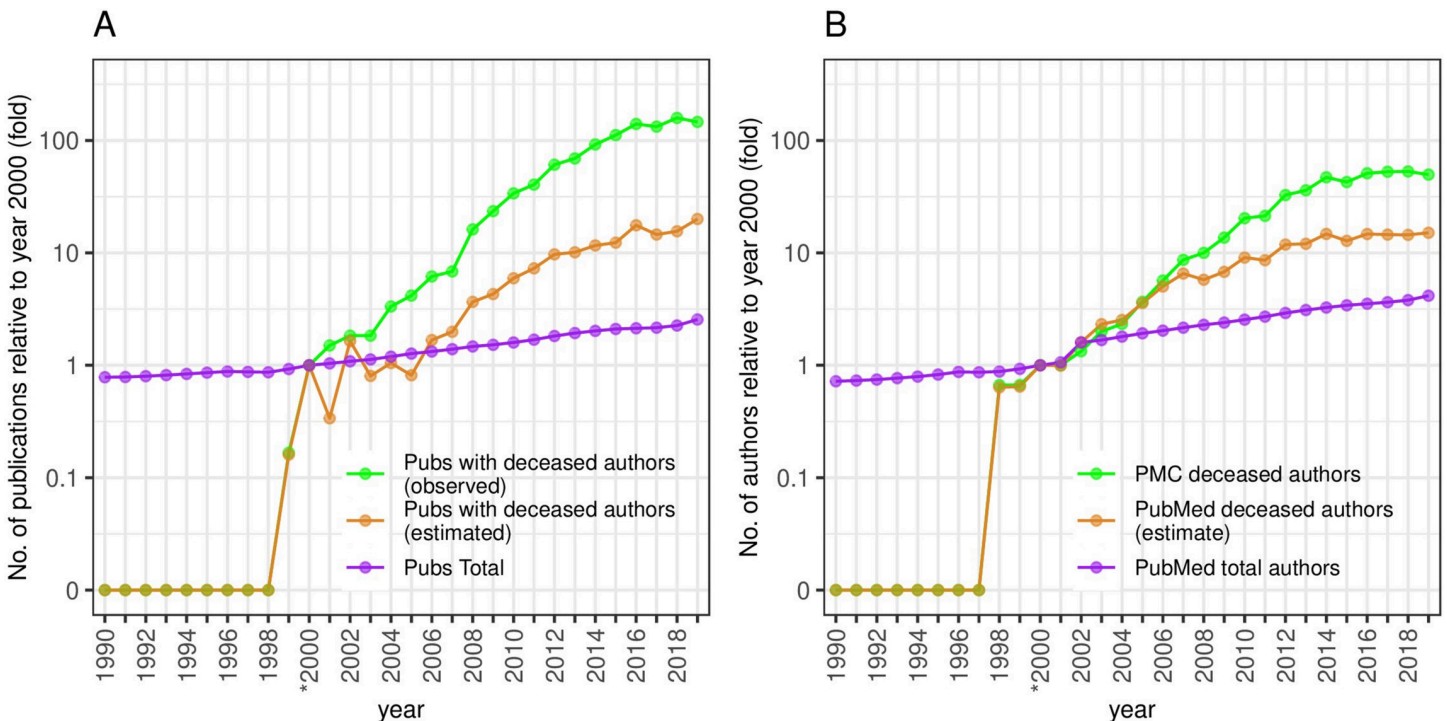

**Fig 1. Rapid growth of publications with deceased authors in the PubMed database.** (A) The fold increase of the relative increase of publications containing a deceased author (green) and the relative growth of total publications (purple) and the adjusted estimated deceased-author publications (orange) in PubMed. The baseline for deceased author and total publications was set at year 2000 (*). (B) Rapid growth of deceased authors in the EPMC database. The growth of deceased authors in the biomedical literature (green), the growth of the pool of total unique authors in PubMed (purple) and the estimated number of deceased authors in PubMed (orange). The baseline for deceased and total authors was set at year 2000 (*).

increase in the pool of authors, and a concomitant increase in the pool of deceased authors. We estimated the pool of unique authors publishing each year from the PubMed database. Total authors grew 4.2-fold over this period, corresponding to an annual growth rate of 7.8%. The number of deceased authors in PubMed, which was estimated by the EPMC author-death rate, grew 15-fold, corresponding to an annual growth rate of 15.5%. By contrast, the increase in deceased authors found in EPMC was ~50 fold over this period, an annual growth rate of 23.5% (Fig 1B). Thus, the increase in publications by deceased authors is not fully explained by a growing total number of publications nor of a growing pool of authors.

To quantify the extent to which the observed increase in publications by deceased authors exceeds the expected rate we developed a regression model. We modelled the number of deceased author publications (dp) as a function of total PubMed publications (*p*), total unique PubMed authors (*a*) and estimated author death-rate [18] (*d*) for each year: $dp = f(p, a, d)$. We fit this multiple linear regression model over the three, six-year periods, 1999–2004, 2004–2009 or 2009–2014, to predict the number of PubMed publications with deceased authors in the remaining years to 2019, which incorporate into the models the changing rate of growth over these time periods. The observed increase in publications by deceased authors significantly exceeds the expected rate of such publications regardless of the model (Fig 2A).

The third criterion of the ICMJE guidelines states that all authors must have approved the final accepted manuscript and it is possible that an author may die after a paper is accepted by a journal and therefore still fulfil this criterion. To determine how many publications in our collected data could not fulfill this criterion we calculated the number of papers where the first submission date occurred after our conservative estimate of the date on which the deceased

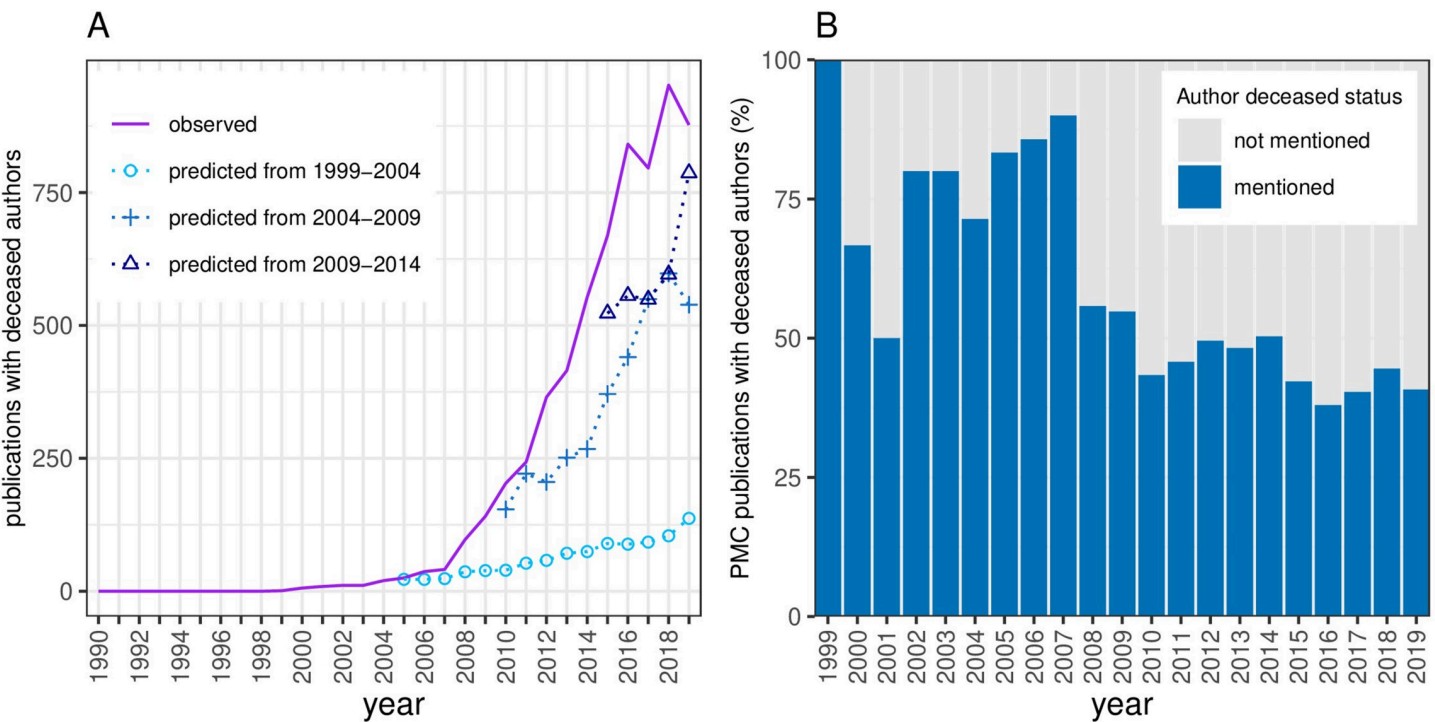

**Fig 2. Regression model.** (A) Projection of the number of publications with deceased authors based on the trend from three different time intervals; 1999–2004, 2004–2009, 2009–2014 The Predicted trends (dashed blue lines) versus the actual Observed trend (purple line). (B) Percentage of publications (EPMC) with deceased authors with mention of the deceased author in the Acknowledgements. Dark blue bars represent the percentage of publications with deceased authors mentioned in the Acknowledgement and stacked grey bars represent deceased author publications with no mention of the deceased author in the Acknowledgements.

author passed away. From the 3,955 papers with a deceased author with submission date information, 53% (2,113) were first submitted after the date of death. Journals vary in policies around annotation of submission date, particularly for papers with long revision processes, making these values tentative.

## Reporting of deceased authors in publications

Regardless of whether a particular journal adopts the ICMJE criterion or not, most journals that have a policy regarding deceased authors request that the deceased author's status is made clear in the publication. We checked how many publications with deceased authors and full text available had a reference to a deceased author in the acknowledgements. Amongst the 3,850 papers with a deceased author and full text available, 45% (1,701) mention the deceased author in the Acknowledgements (Fig 2B). Presumably, many of the remaining papers with deceased authors might include references in other parts of the manuscript. Nevertheless, this reflects significant variability in current reporting practices.

Of those authors classified as deceased we then assessed what fraction of their total publication output was represented by their deceased publications. We observed that for the vast majority of deceased authors, most have published 10 or less papers after death (Fig 3). There is however an outlier group of >30 authors who have published a total of more than 20 papers after their deceased date with a small coterie of deceased authors publishing more than 50 papers after death with the highest having over 165 papers as an author after their death date, spanning a period of some 9 years. For most of the deceased authors, the deceased-publications account for 50% or less of the total publications. However, for 266 deceased authors (18.5% of the total), the proportion of their deceased-publications among their whole publications is 90% or more (Fig 3).

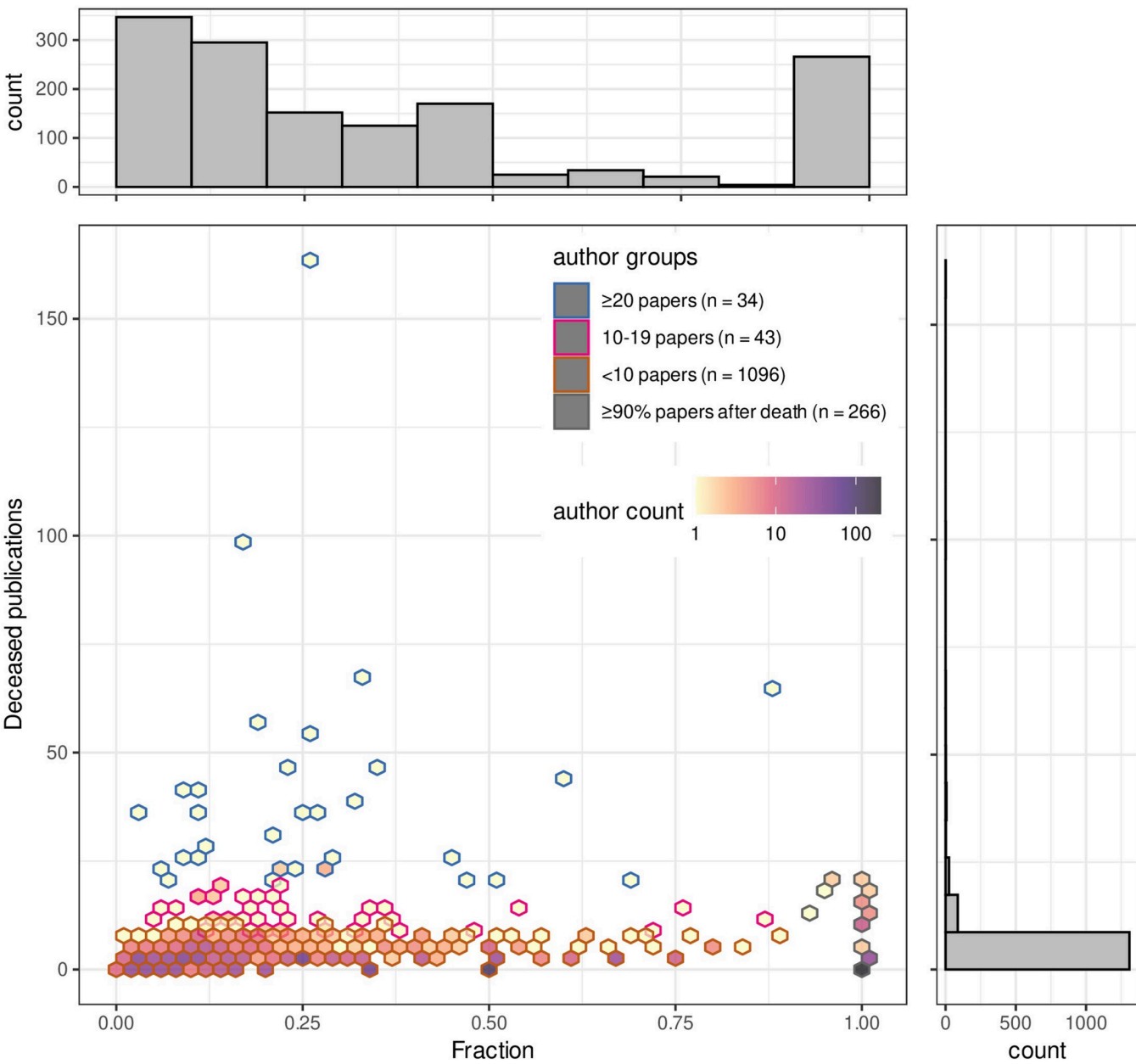

**Fig 3. Fraction and the raw number of publications after death.** Each hex bin dot represents a deceased author. The fraction of their deceased-publications compared to their total publications are on the X-axis and the raw number of their deceased-publications on the Y-axis. The fraction of deceased-publications was calculated from the total publications published by deceased authors over their careers. Authors were grouped into four based on the number of publications after death and the fraction of deceased-publications over the career. Either; authors with 20 or more deceased-publication; authors with 10 to 19 deceased-publications; authors with less than 10 deceased-publications, or authors who published 90% or more of the total publication as deceased-publications regardless of the number of deceased-publications. Histogram plots represent the actual number of either the fraction of deceased publications (top) or number of deceased publications (side). Hex bin dot colors refer to numbers of deceased authors across a continuous range as depicted.

## Trends in deceased author publications

There appears to be a generalized trend for deceased authors to have an increasing number of annual publications published until the year of death, which then decreases rapidly after the year of death. The observed increasing rate of publication followed by the subsequent decline in the number of publications within an 11-year period (from 5 years-prior to 5 years-post

from the year of death) was particularly pronounced for those authors with 10 or more deceased publications (Fig 4A). For the two top-publishing author groups, namely those ≥ 20 deceased publications and those with 10–19, the volume of publications for these authors increased at a similar rate with a subsequent rapid decline post their deceased date. In contrast, the majority of the authors in the other two groups comprising those with < 10 deceased publications published only a small number of papers consistently until the year of death and then virtually ceased publishing after one or two years after death (Fig 4A). This similar trend was also observed in the proportion of publications over their career within each year of the 11-year period (not shown). The number of co-authors publishing with a deceased author, could possibly influence the number of publications that a deceased author accrues after death. A larger pool of living potential co-authors could increase the probability of extra publications after death. We therefore checked the number of co-authors of publications with deceased authors, by observing the trends for all deceased authors from the period of 5 years prior to death until 5 years after death. For all deceased authors, regardless of their grouping, the author-team size gradually increased upto the year of death, and then continue to rise after the year of death, however this was most marked for the most prolific deceased author group, i.e. those with more than 20 deceased publications (Fig 4B). The author-team size trend increase persists after the year of death and tends to peak at the third or the fourth year after death before starting to fall (Fig 4B).

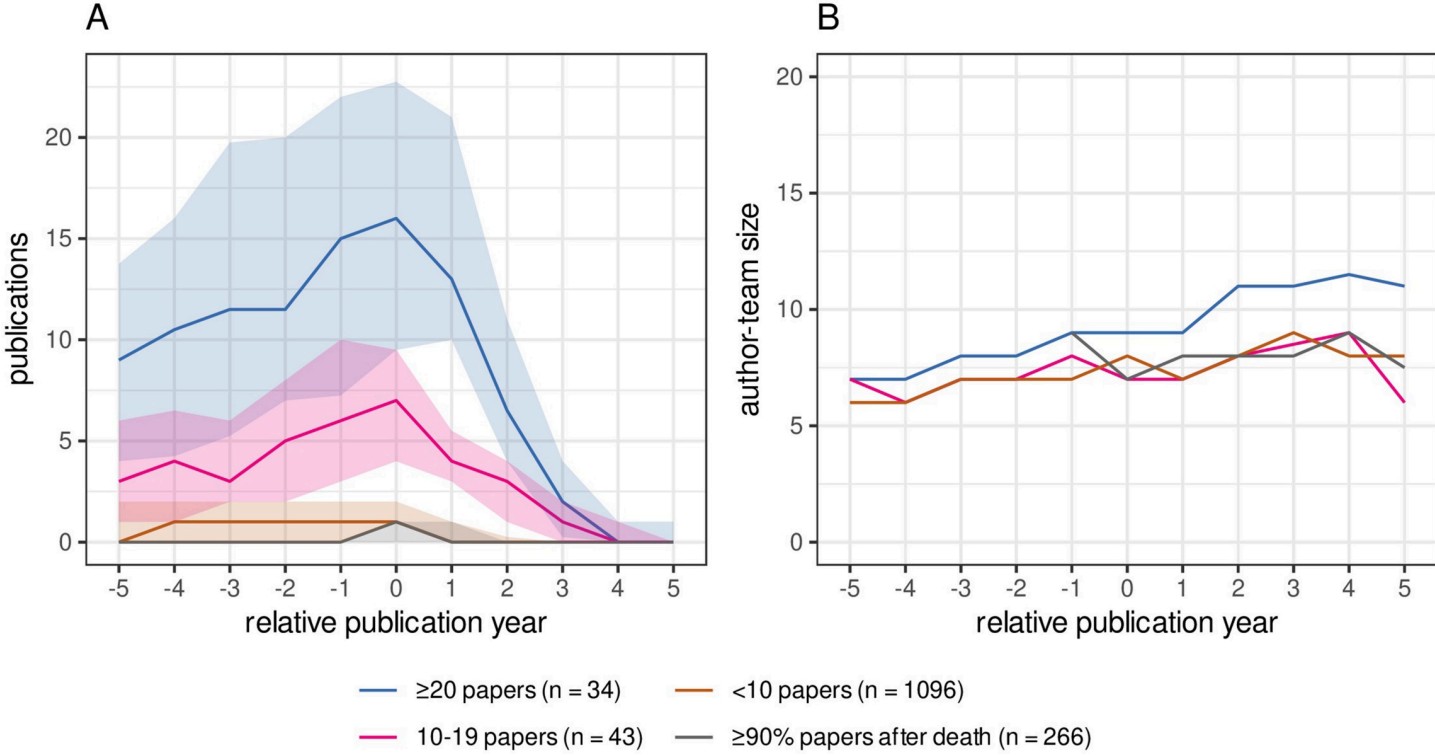

**Fig 4. Trend of publication counts around the year of death.** (A) The number of publications (y axis) by deceased authors by deceased author publication groupings over an 11-year time span encompassing 5 years before death to 5 years after year of death (x axis). The lines represent the median, and the shaded area represents the interquartile range. (B) Author-team size of publications by deceased authors. The number of co-authors for each publication with a deceased author was calculated and depicted over an 11-year span. Lines represent the median of the author-team size of publications within 5 years before and after the year of death and the shades represent the interquartile range. Colors depict four different groupings of deceased authors based on the number of their deceased publications ranging from those with ≥ 20 such publications (orange) and those with ≥ 90% of their total publications after death (grey).

We performed a word cloud analysis of MeSH (Medical Subject Headings) terms from the approximately 5,000 deceased author publications which revealed that papers whose topics encompassed biomedical molecular analyses featured heavily in the MESH topics of these papers (Fig 5A). This is perhaps reflecting a propensity for cross border collaborative studies in these fields also reflected in the larger author teams on these papers compared to randomly selected comparison cohorts (not shown). We then plotted the country affiliations of the first authors of all publications with deceased authors, focusing on the top 11 countries to observe whether deceased author publications were evenly distributed across the cohort. We also compared the number of deceased publications compared to the number of before death publications for all deceased authors to observe if prior to death publications affected the rate of post death publications. A clear trend emerges where the number of post-death publications is strongly correlated to the number of prior death publications by those authors, with the authors from the United States having by far the largest number of pre- and post-death publications (Fig 5B).

To observe the timescale of deceased author publications in relation to the year of death of these individuals we plotted the total cumulative posthumous publications across a 16 year time scale around the year of death for the whole cohort of deceased authors, which represents the longest span of time we have found papers published by deceased authors. We observed that the trend we noted for the most prolific deceased authors was maintained across the whole cohort of deceased author publications (Fig 6). A steady increase in the numbers of annual publications from deceased authors prior to their deaths was apparent, peaking at the year of death. There was then a subsequent marked annual decline in the number of publications, stretching however to 11 years post the actual death of authors (Fig 6A). This trend was also observed for the numbers of deceased authors with a slower but steady increase in the number of deceased authors culminating in a marked peak at the year of death which then

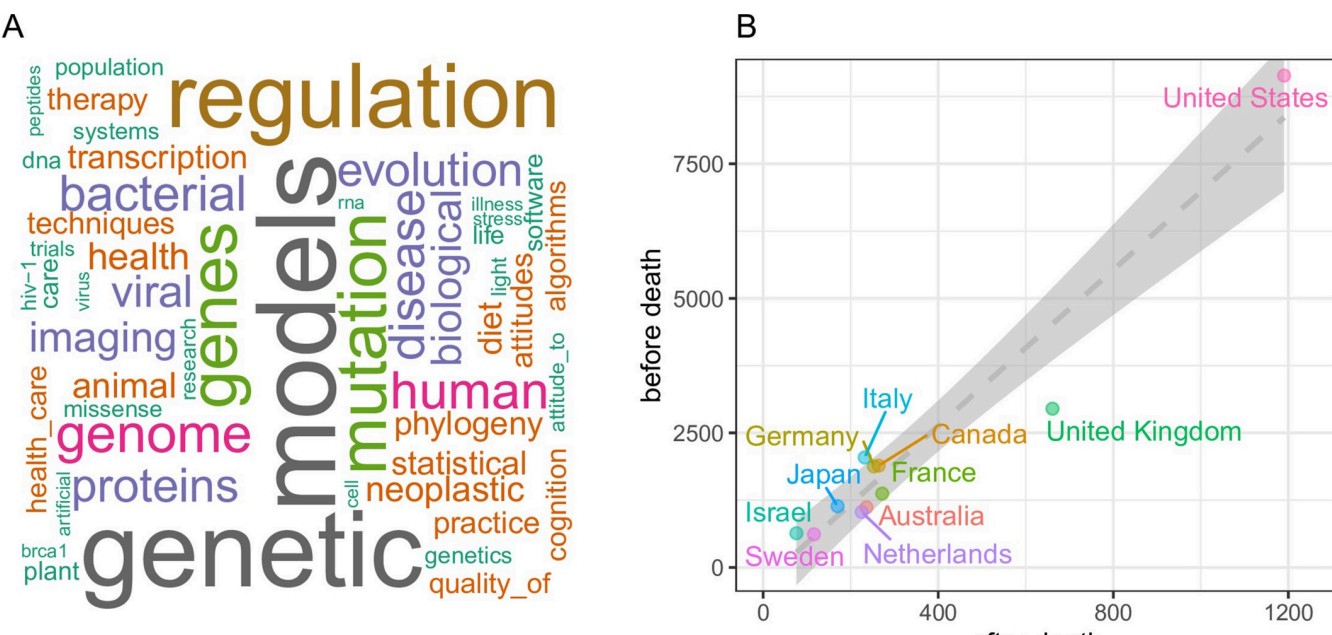

**Fig 5.** Word cloud representations of (A) topic MeSH terms and (B) country affiliations of first authors of deceased publications. (A) The word cloud image is a visual representation of word frequency derived from analysis of the full text of the deceased author publications in the database. The more often the word appears within the text the larger it's appearance in the image. (B) Plot of first author country affiliations of publications containing a deceased author comparing after death publications on the X-axis versus before death publications Y-axis. Only the top 11 countries are depicted.

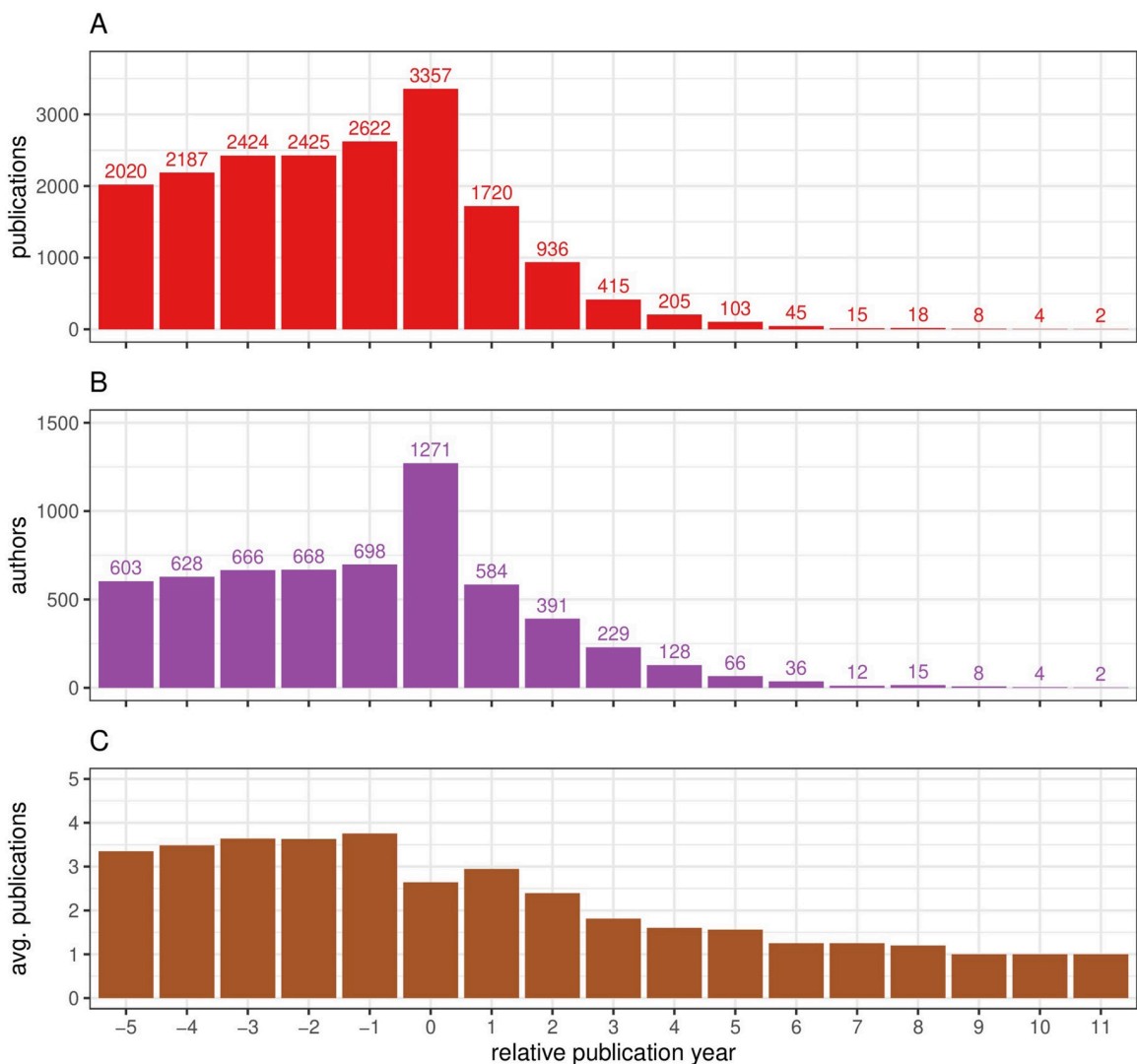

**Fig 6. The timeline of publications by deceased authors.** The span of cumulative publications by the total pool of the deceased author cohort (n = 1439) from 5 years prior to the year of death till 11 years after the year of death is depicted in (A) red bars and the numbers of authors (B) in blue, with average publications per author (C) in brown. The count represents actual numbers of both publications and deceased authors for each year.

rapidly declined after death (Fig 6B). The average publications by the deceased authors before their death increased slightly from 3.3 to 3.8 between 5 and 1 years prior to death. The average of publications by deceased authors then decreased at a slower rate than the raw numbers of either authors or publications suggesting that the relative productivity of deceased authors remains relatively stable upto 11 years post the year of death (Fig 6C).

## Discussion

Our literature-wide analysis of the extent of deceased author publications has revealed several surprising findings. Firstly, the rate of such publications in the literature has been growing rapidly, and at much higher rates than can be accounted for by confounding factors such as the annual increases in total publications and the concomitant increase in the pool of authors. These findings were confirmed by the regression analysis over multiple time periods indicating

that the observed rate of increase cannot be adequately explained by known factors likely to influence this rate. We restricted our analysis to the 30-year period of 1990–2020 but could find no publications in the full text EPMC database with terms referencing a deceased author in the Acknowledgment section prior to 1999. This suggests that the occurrence of deceased authorship in the biomedical literature prior to these dates was likely sporadic.

We also found that more than 50% of deceased author papers were first submitted after the death of the deceased co-author and that over 60% of these papers failed to acknowledge the deceased authors status. These findings, particularly that of first submission dates prior to the death of a co-author would make compliance with current ICMJE criteria justifying inclusion as an author, problematic. Of the 1,439 deceased authors we identified in the literature, the majority published less than 10 publications as deceased authors. However, two outlier groups of these authors warrant further comment. Firstly, a large cohort of 266 deceased authors (18.5% of the total) appeared to have published more than 90% of their total publications after death, including a large number that apparently only published in the literature after their deaths. These findings are however tentative, as the uncertainty in ascribing a given author to a publication is greater prior to their death. This is because both affiliations and author names can change throughout their careers making our estimates of prior death publications likely lower than the real figure.

A small cohort of approximately 30 deceased authors published more than 20 papers after their deaths, with one author reaching a current maximum of 165 papers. Whilst the number of these authors is small overall, their rate of productivity would seem to challenge current definitions of appropriate criteria for authorship. We found that the most prolific deceased authors, those with more than 10 such publications, were also the most productive in the 5-year period prior to their deaths and hence the rate of their post-death publications is likely a reflection of these authors overall research and collaborative activity whilst alive.

The distribution of deceased author publications across different journals is widespread, though our scaled frequency analysis indicates that some journals appear to publish more of these publications than expected (S1-S3 Figs in S1 File). Surprisingly, our odds ratio analysis indicated that for two journals, *Scientific Reports* and *Nature Communications*, a deceased author had a higher likelihood of publishing in these journals after death than before (S3 Fig in S1 File). However, the reason for this was not apparent, as both journals have authorship criteria equivalent to other journals.

Authorship plays a central role in the ways credit is assigned in biomedicine [4,20,21]. As a result, its definitions and uses have been widely discussed, with frameworks codified [22–25]. If ICMJE criteria were strictly followed, it would be impossible for most deceased authors to fulfil all four, particularly providing final approval of the manuscript [26]. The timing of the author's passing would be relevant: agreeing to be accountable for all aspects of the work might be considered feasible if the study was fundamentally completed by that date. Some journals now provide guidelines regarding inclusion of deceased individuals as authors. The *Journal of the American Chemical Society*, *BMJ Journal*s, *Pediatric Anesthesia*, the Cochrane Community and the Council of Science Editors have modified criteria regarding author inclusion, in some cases relaxing ICMJE criteria. Perhaps paradoxically, *BMJ Journals* maintain strict ICMJE criteria but still permit deceased persons as co-authors by requiring a footnote reporting the date of death [15,27].

This analysis substantially underestimates the extent of publication by deceased authors in the biomedical literature. We restricted our search terms to only full text acknowledgement sections of papers in the Europe PMC database. Further, we rely on co-authors to report the presence of a deceased author. These restrictions may also cause us to underestimate the true extent of deceased author publications in the literature prior to 1999. Prior to this, deceased

authors may have been acknowledged in alternative sections of a paper. We analyzed a subset of the papers around the 1999 period for alternative listings of deceased author status, including funding acknowledgements, but could find no such attributions.

Given the lack of standard guidelines, co-author interpretation of appropriate reporting practice may be variable. Indeed, one noteworthy explanation for the observed increase in deceased-author publications is an evolving community understanding of what warrants authorship in scientific publications [28,29].

Authorship guidelines are essential to reduce instances of fraudulent and honorary authorship. It is possible that honorary authorship could be conferred on deceased individuals with a high profile and established reputation to improve a study's impact [30,31]. The increasing contribution of deceased authors in the biomedical literature could also exacerbate the growing trend in average number of authors per publication which also poses an ethical issue for authorship in general [32]. However, improper motives may be less likely to be at play in the majority of cases of deceased authors as the majority do not have significant publication records and further they cannot exert undue pressure or coerce co-authors for honorary authorship after death. Rather, it is more likely that living co-authors recognize the important contribution of their deceased colleagues and genuinely wish to attribute credit for this. This would suggest that many co-authors believe that fulfilling of ICMJE authorship criteria 1, namely substantial contribution to a scholarly work, is sufficient to warrant authorship. Some journals are now modifying the ICMJE criteria as to what qualifies for authorship and this follows calls in the literature to update the strict ICMJE criteria [33,34]. There has been a growing realization that conventional author attributions are increasingly outdated and fail to distinguish amongst the wide range of potential contributions that individuals can make to a scholarly work [35,36]. The CRediT project has developed a controlled vocabulary of contributor roles, in effect a taxonomy, for publications in biomedicine (https://credit.niso.org/) and this taxonomy is increasingly being adopted by journals and publishers to aid in transparency and reproducibility in research. A combination of clear and correct attribution of a deceased author status in publications such as requested by the *BMJ journals* along with use of the CRediT taxonomy to clearly describe each individual's specific contribution to a published work, would potentially remove much of the controversy around the role of deceased individuals in the biomedical literature.

Our analysis indicates that the attribution of deceased authorship in the literature is no longer an occasional occurrence but in fact is a burgeoning trend. It is also wholly unclear to what extent the wishes of the deceased biomedical scientists are being met. A consistent, considered consensus framework to address authorship by deceased biomedical scientists would appear to now be warranted.

## Conclusion

In summary, our study identified a minimum set of 1,439 deceased authors who published in the biomedical literature between 1990–2020. These authors published 5,477 papers after death. Relatively few of these were published between 1990–2000 but since 2000 there has been a 146-fold increase. Over 50% of these papers were first submitted after the death of the author and more than 60% of these papers failed to acknowledge the deceased authors status. Most deceased authors published less than 10 papers after death but a small pool of 30 authors published significantly more. A pool of 266 authors published more than 90% of their total publications after death. A consensus framework to address the issue of authorship by deceased individuals is warranted.

## Supporting information

**S1 Table.**
(XLSX)

**S1 File.**
(PDF)

## Acknowledgments

We thank Professor Tony Papenfuss from the Walter Eliza Hall Institute and Dr. Anis Hamid from the University of Melbourne for helpful discussions during the early phase of this research.

## Author Contributions

**Conceptualization:** Christopher M. Hovens.

**Data curation:** Chol-Hee Jung, Paul C. Boutros, Bernard J. Pope, Christopher M. Hovens.

**Formal analysis:** Chol-Hee Jung, Bernard J. Pope, Christopher M. Hovens.

**Funding acquisition:** Christopher M. Hovens.

**Investigation:** Chol-Hee Jung, Paul C. Boutros, Bernard J. Pope, Christopher M. Hovens.

**Methodology:** Chol-Hee Jung, Paul C. Boutros, Bernard J. Pope, Christopher M. Hovens.

**Project administration:** Christopher M. Hovens.

**Resources:** Bernard J. Pope, Christopher M. Hovens.

**Software:** Chol-Hee Jung.

**Supervision:** Bernard J. Pope, Christopher M. Hovens.

**Validation:** Chol-Hee Jung, Bernard J. Pope, Christopher M. Hovens.

**Visualization:** Chol-Hee Jung, Paul C. Boutros.

**Writing – original draft:** Chol-Hee Jung, Paul C. Boutros, Christopher M. Hovens.

**Writing – review & editing:** Chol-Hee Jung, Paul C. Boutros, Daniel J. Park, Niall M. Corcoran, Bernard J. Pope, Christopher M. Hovens.

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
