## [Decision Letter · Decision Letter 0]

17 Apr 2022

PONE-D-21-35578Perish and Publish: Dynamics of Biomedical Publications by Deceased AuthorsPLOS ONE

Dear Dr. Hovens,

Thank you for submitting your manuscript to PLOS ONE. After careful consideration, we feel that it has merit but does not fully meet PLOS ONE’s publication criteria as it currently stands. Therefore, we invite you to submit a revised version of the manuscript that addresses the points raised during the review process. Two experts on publication practices and policies reviewed your submission. Both the reviewers and I agree that that the issue you raise is relevant and that the data collection and analyses were clearly described and that your conclusions are sufficiently supported by the data. However, several issues need to be addressed in your revision. First, Reviewer 1 rightly indicated that you present a substantial number of supplementary results and I agree that many of these results in later sections of the manuscript are less relevant, and could be placed in an (online) appendix. So a major revision would shorten the paper to present the main results but not the supplementary results with respect to journals. Second, I also agree with the second reviewer that you performed many exploratory tests to document particular patterns but that these appeared not to be based on any prior hypotheses. It is important to clearly identify tests that were data driven from the key tests about which you had prior hypotheses and to identify the former types of tests as exploratory and in need of fresh data to ensure replicability and robustness. Third, you need to devote more attention to the main limitations of the database and data collection and be open and critical on what they might mean. Keep in mind that we at PLOS ONE value rigor over "newsworthiness" of the findings and so I urge you to be openly and meticulously discuss methodological issues such as those mentioned by the second reviewer. Here, I also noted the dramatic increase from null occurrences before 1999 and afterwards that look to me like they could be driven by changes in text section coverage or other factors that would be good to consider (e.g., by checking trends in funding statements as a control). Fourth, the second reviewer raises specific issues related to the ICMJE definition and improper motives to include a deceased author and other issues that you need to carefully consider when revising your work.

We look forward to receiving your revised manuscript.

Kind regards,

Jelte M. Wicherts

Academic Editor

PLOS ONE

Journal Requirements:

3. Please update your submission to use the PLOS LaTeX template. The template and more information on our requirements for LaTeX submissions can be found at http://journals.plos.org/plosone/s/latex

Reviewers' comments:

Reviewer's Responses to Questions

**Comments to the Author**

1. Is the manuscript technically sound, and do the data support the conclusions?

Reviewer #1: Yes

Reviewer #2: Yes

2. Has the statistical analysis been performed appropriately and rigorously? 

Reviewer #1: Yes

Reviewer #2: I Don't Know

3. Have the authors made all data underlying the findings in their manuscript fully available?

Reviewer #1: Yes

Reviewer #2: Yes

4. Is the manuscript presented in an intelligible fashion and written in standard English?

Reviewer #1: Yes

Reviewer #2: Yes

5. Review Comments to the Author

Reviewer #1: To provide some empirical food to the normative discussion of whether deceased researchers should be included as author - how large a problem is this is in practice? - the authors have explored a large number of publications in the Europe Pubmed Central database. Outcomes are thoroughly reported.

Reviewer #2: This paper explores the prevalence of posthumous authorship in biomedical literature. The paper is well-written with a clear beginning and ending but it is not ready for publication yet. Below are my specific comments per section:

Introduction:

- The current introduction is quite superficial. Authors need much better contact with the available literature about posthumous authorship.

- References 1-8 discuss a range of different issues and citing all of them after a generic claim is 1) a cop out, 2) an example of citation padding.

- References 11-15 too, discuss a range of different issues and citing all of them in one cluster is 1) a cop out, 2) an example of citation padding.

- When relying on the ICMJE definition, please make sure that the definition of authorship is accurately reflected. The ICMJE definition does NOT require "intellectual" contributions, it requires "substantial contributions". Also, when referring to the ICMJ, please avoid citing secondary sources that interpret the definition.

Results:

- There's a good bit of repetition in this section.

- I suggest that you use sub-headings to improve readability.

- Authors note that some journals have a policy regarding posthumous authorship. This should be added to the introduction with some examples.

- In most of the provided analyses, a hypothesis is missing. Using "we wondered" is not necessarily helping because some could interpret this as an example of HARKING. Authors use a range of different statistical methods without clarifying the link between their hypothesis and used method.

Discussion:

- Please make sure that all results are included in the results section and methods are in the methodology. Currently some of the results and methods are in the discussion section.

- References 20-27, 1) a cop out, 2) an example of citation padding.

- Authors note "improper motives may be less likely at play in the case of deceased authors because they

cannot exert undue pressure or coerce co-authors for honorary authorship after death", but this is actually not true. e.g., there are plenty of cases wherein junior authors added senior authors to their papers without seniors' consent, which resulted in authorship disputes and complaints by seniors. In such cases, senior authors are added with the hope to increase impact/visibility and exhibit good connections etc. One can imagine that in posthumous authorship too, some groups would try to link themselves with high-profile researchers to improve their study's impact etc.

6. PLOS authors have the option to publish the peer review history of their article (what does this mean?). If published, this will include your full peer review and any attached files.

Reviewer #1: No

Reviewer #2: **Yes: **Mohammad Hosseini

---

## [Author Response · Author response to Decision Letter 0]

25 May 2022

we have uploaded a separate response to reviers letter that addresses all the issues raised.

---

## [Decision Letter · Decision Letter 1]

7 Jul 2022

PONE-D-21-35578R1Perish and Publish: Dynamics of Biomedical Publications by Deceased AuthorsPLOS ONE

Dear Dr. Hovens,

Thank you for submitting your revised manuscript to PLOS ONE. After careful consideration, we feel that it has merit but does not fully meet PLOS ONE’s publication criteria as it currently stands. Therefore, we invite you to submit a revised version of the manuscript that addresses the points raised during the review process.

The second reviewer from the previous round  and I considered your revisions and although we agree that your revisions improved your submission, several issues remain. I agree with the reviewer that the text needs revision and a restructuring such that the methods are part of the main text rather than put at the end. The reviewer also makes specific points concerning the text, some inaccuracies, methodological issues, and earlier work in this area that you can readily address. I also noted that the text needs some further editing (note that PLOS ONE does not offer text editing to lower the APC) and another check of minor inaccuracies and ambiguities in the text. If you deal well with these issues, I might be able to make a final decision on the manuscript without sending it out for review again. 

We look forward to receiving your revised manuscript.

Kind regards,

Jelte M. Wicherts

Academic Editor

PLOS ONE

Journal Requirements:

Reviewers' comments:

Reviewer's Responses to Questions

**Comments to the Author**

1. If the authors have adequately addressed your comments raised in a previous round of review and you feel that this manuscript is now acceptable for publication, you may indicate that here to bypass the “Comments to the Author” section, enter your conflict of interest statement in the “Confidential to Editor” section, and submit your "Accept" recommendation.

Reviewer #2: (No Response)

2. Is the manuscript technically sound, and do the data support the conclusions?

Reviewer #2: Partly

3. Has the statistical analysis been performed appropriately and rigorously? 

Reviewer #2: I Don't Know

4. Have the authors made all data underlying the findings in their manuscript fully available?

Reviewer #2: No

5. Is the manuscript presented in an intelligible fashion and written in standard English?

Reviewer #2: No

6. Review Comments to the Author

Reviewer #2: I appreciate authors' efforts in responding to the previous comments and can see that the paper is improving but it is still not ready for publication because there are areas that the paper could improve. Please note that I am trying to help you here because I think this is an important paper that should be published in an outstanding shape. I suggest authors to spend some time for a careful reading and revising of the text. In what follows, I provide some comments:

Please remove citation nr. 5, when using direct quotations, you cannot cite two independent sources!

Authors note “For example, if an author passes away between formal acceptance of a peer-reviewed paper and its publication, there may be robust documentation of them meeting all four criteria.”, which negates the opening sentence of the exact same paragraph: “if these criteria were followed strictly it would be impossible for any deceased author to fulfill all four criteria.”

Authors note “Some biomedical journals, such as, BMJ Journals, Pediatric Anesthesia have now developed their own policies with respect to deceased authors, potentially fragmenting the consensus understanding of what constitutes authorship.”, I suggest authors to provide further explanation on some details of these policies and a clear explanation about why these policies fragment the consensus understanding. Furthermore, creating a “consensus understanding” is vague: “consensus among who?”

Also, what is meant by consensus understanding? If, as authors claim in their opening statement, “The concept and definitions of what constitutes authorship in scientific publications has been much commented in the literature and numerous frameworks…”, there is no consensus about authorship to begin with! Creating a consensus about any authorship issue is almost impossible, so this is s flimsy rational anyway. I’d suggest using “consistent practice in the community”, “harmonized practice” or something similar so that you don’t imply that creating consensus is actually possible.

A lot of what is currently mentioned in the results section especially what is under “Compiling a database of posthumous authors” should be in the “Methodology” section, which is currently after the acknowledgement section! PLease make sure that the paper has a solid structure and flow. At the moment, the paper meanders a good bit.

A lot of what is currently mentioned in the results section especially what is under “The cumulative change in deceased author publications over time” should be mentiond in the Intro, as the rationale for the study. Furthermore, another rationale could be that adding dead authors increases the average number of authors per publication, which has been reported to exacerbate all ethical issues of authorship at some level (https://link.springer.com/article/10.1007/s11948-021-00352-3)

The sentence “It is possible that some cases of posthumous authorship in our analysis include the use of a high profile researchers profile to improve a study’s impact” is badly written

Refernce #26, Kosmulski 2021 has some really useful content for this paper. I suggest authors to carefully read and add some recommendations to their own paper. For instance, authors could highlight the fact that "Neither HTLM not PDF versions on journals' Web pages inform that one of the authors was deceased" and suggest that publishers should be more proactive and transparent in declaring whether the author was dead at the time of submission, during the review process, between acceptance and publication etc.

7. PLOS authors have the option to publish the peer review history of their article (what does this mean?). If published, this will include your full peer review and any attached files.

Reviewer #2: No

---

## [Author Response · Author response to Decision Letter 1]

9 Aug 2022

we have uploaded separately a new response to reviewers file.

---

## [Editor Report · Decision Letter 2]

16 Aug 2022

Perish and Publish: Dynamics of Biomedical Publications by Deceased Authors

PONE-D-21-35578R2

Dear Dr. Hovens,

We’re pleased to inform you that your manuscript has been judged scientifically suitable for publication and will be formally accepted for publication once it meets all outstanding technical requirements. I don't see any ethical problems with you sharing the names of the deceased authors as an appendix.

Kind regards,

Jelte M. Wicherts

Academic Editor

PLOS ONE
---

## [Editor Report · Acceptance letter]

23 Aug 2022

PONE-D-21-35578R2 

Perish and Publish: Dynamics of Biomedical Publications by Deceased Authors 

Dear Dr. Hovens:

I'm pleased to inform you that your manuscript has been deemed suitable for publication in PLOS ONE. Congratulations! Your manuscript is now with our production department. 

Kind regards, 

on behalf of

Dr. Jelte M. Wicherts 

Academic Editor

PLOS ONE